# Effect of Anthocyanin-Enriched Brine on Nutritional, Functional and Sensory Properties of Pickled Baby Corn

**DOI:** 10.3390/plants12091812

**Published:** 2023-04-28

**Authors:** Marijana Simić, Valentina Nikolić, Dubravka Škrobot, Jelena Srdić, Vesna Perić, Saša Despotović, Slađana Žilić

**Affiliations:** 1Department of Food Technology and Biochemistry, Maize Research Institute, Zemun Polje, Slobodana Bajića 1, 11185 Belgrade, Serbia; valentinas@mrizp.rs (V.N.); szilic@mrizp.rs (S.Ž.); 2Institute of Food Technology, University of Novi Sad, Bulevar cara Lazara 1, 21000 Novi Sad, Serbia; dubravka.skrobot@fins.uns.ac.rs; 3Plant Breeding Department, Maize Research Institute, Zemun Polje, Slobodana Bajića 1, 11185 Belgrade, Serbia; jsrdic@mrizp.rs (J.S.); vperic@mrizp.rs (V.P.); 4Faculty of Agriculture, University of Belgrade, Nemanjina 6, 11080 Belgrade, Serbia; sdespot@agrif.bg.ac.rs

**Keywords:** baby corn product, corn genotypes, black soybean, anthocyanins, phenolic compounds, sensory properties

## Abstract

Considering the great potential of black soybean seed coat as a source of bioactive compounds, the objective of this study was to investigate the effect of anthocyanin-rich brine from the seed coat on functional properties of pickled baby corn, as well as its sensory properties. Given that the ears of sweet corn, popping corn and semi-flint corn were used for pickling in the pre-pollination phase, the effect of genotype and its growing stage on the chemical composition of Baby corn product was also taken into consideration. The brine of black soybean with a total anthocyanins content of 11,882.9 mg CGE/kg (cyanidin 3-glucoside equivalent) and an antioxidant capacity of 399.5 mmol Trolox Eq/kg determined by QUENCHER method had a positive impact on the functional potential of baby corn products. The content of total anthocyanins in the obtained products ranged from 748.6 to 881.2 mg CGE/kg, the predominant anthocyanin was cyanidin-3-glucoside (184.6 to 247.5 μg/g), while their colour was red. Compared to the commercial sample, baby corn products pickled in the enriched solution had a 26% to 46% and 17% to 26% higher content of total free phenolic compounds and antioxidant capacity, respectively. Contrarily, the control sample had higher sugar and fibre content. As established, pickled popping corn had the best sensory properties.

## 1. Introduction

Corn (*Zea mays* L.) is one of the oldest and most widely grown cereal crops that can be consumed as food at various developmental stages, ranging from baby corn to mature grain [1]. The term baby corn refers to the unfertilised cob of corn preferably harvested within two to four days of silk emergence, depending on the growing season. Historically used as a vegetable in China and other Asian countries, baby corn has lately garnered favour on a global scale. This eco-friendly food has a lot of promise as a value-adding product. Baby corn is an excellent appetizing supplement to many traditional and international dishes because of its great digestibility, sweet flavour, softness and crunchiness, as well as its eye-catching colour [2]. Baby corn is highly nutritive, very palatable, rich in dietary fibre, and low in calories and cholesterol [3]. Although in the fully mature stage the corn grains can be intensely orange, red, blue, and even black in colour, baby corn is mostly white to light yellow. In the phase of cob development before fertilization, the synthesis of bioactive compounds that affect the grain’s final colour has not yet taken place. Coloured cereals are primarily anthocyanin-pigmented. However, studies have shown that the content of anthocyanins in sweet corn grain increases significantly only between 20 and 30 days after pollination, which is much later than the development phase in which young cobs are harvested as baby corn [4].

In the food industry, baby corn is mainly subjected to a pickling process that is considered as one of the earliest methods used for food preservation through fermentation [5]. To create products known as pickles, baby corn can be preserved naturally by a controlled fermentation by adding vinegar directly to a solution with a pH set to 4.6 or below, or by a combination of techniques, processing parameters, and additives [6]. Pickled vegetables are good sources of natural antioxidants such as vitamins, carotenoids, flavonoids and other phenolic compounds [7,8]. However, changes in the everyday diet have made the bioactive component-enriched foods increasingly popular [9]. There is great potential for value adding on baby corn products with red beet [10], as well as with the marinating brine rich in anthocyanins extracted from black soybean seed coat. By adding anthocyanins derived from black soybean seed coats, canned corn products such as sweet corn and baby corn can have their nutritional and health-promoting qualities enhanced [11]. Anthocyanins are the largest group of phenolic pigments, and their in vitro and in vivo studies have demonstrated nutraceutical potential, high antioxidant capacity and many biological effects. The consumption of foods high in anthocyanins has been linked to a wide application in the prevention as well as in the delay of the onset and even in the treatment of various human diseases caused by oxidative stress, such as obesity, diabetes, cancer, degenerative diseases of aging, cardiovascular disease, Alzheimer’s and Parkinson’s disease [12,13]. Several authors reported that among all pigmented soybeans, seed coats of black soybeans contain the greatest levels of anthocyanins [14,15], predominantly cyanidin, delphinidin, and pelargonidin as 3-*O*-glucosides [16]. More than 99% of the total anthocyanins in black soybean seed coats (BSSCs), which largely contribute to the colour, are assumed to be located in the epidermal palisade layer [15]. Many studies have shown that BSSCs rich in polyphenols and dietary fibres can be used as bioactive ingredients in functional foods and pharmaceutical products for targeting different health problems [17]. In addition, black soybean extracts rich in bioactive compounds have so far been utilized in the preparation of noodles [18,19,20], crackers [21], touhua (soft soybean curd) [22] and biodegradable films [23,24].

The objective of this study was to investigate the effect of enrichment of the marinating brine with anthocyanins extracted from black soybean seed coat (BSSC) on the nutritional, functional and sensory properties of the pickled baby corn products in order to create a new functional food product with added value. The effect of corn genotype and its growing stage on the chemical composition of baby corn product was also taken into consideration, given that the ears of sweet corn, popping corn and semi-flint corn were used for pickling in the pre-pollination phase.

## 2. Results and Discussion

### 2.1. Chemical Composition of Baby Corn Samples

The chemical composition of the investigated baby corn samples is presented in Table 1. Baby corn is highly nutritive, as it contains on average 15.6% protein, 6.4% total sugar, 25.4% hemicellulose, and 4.5% cellulose (Table 1). The results are comparable to the findings of Kaur et al. [10], who reported that baby corn is a good source of various nutrients and that its nutritional status is on par or even superior to many other commonly used vegetables. In order to evaluate the chemical composition of enriched baby corn samples, all the results were compared with those of commercial baby corn samples as a control. According to our results, the protein content in baby corn samples ranged from 12.69% (semi-flint baby corn with salt and sugar) to 17.86% (control baby corn), and significant differences (*p* < 0.05) were found among all baby corn samples. Our results are consistent with those of Hooda and Kawatra [25], who reported a protein content of 17.9% in the HM-4 variety of baby corn, as well as those of Sinha and Sharma [26], who found that the protein content range was from 11.53% to 18.5%. Semi-flint baby corn samples with salt had a 15% lower protein content (14.61%) than those of popping and sweet ones (17.21% and 17.08%, respectively) as well as 18% lower than those of control baby corn (17.86%). The addition of sugar increased these differences, so the semi-flint baby corn sample had 27.8%, 22% and 15% lower protein content than that in the control, popping and sweet baby corn, respectively. Different corn varieties’ endosperms differ greatly in terms of the protein matrix thickness, hardness, thickness of the peripheral layers, cell sizes, and protein components [27], which was confirmed by our research as well. However, a previous investigation by Nikolić et al. [28] showed that the protein content in the kernels of different corn hybrids ranged from 8.63 to 10.72%. In addition, the results of Watson [27] indicated that the commodity yellow dent corn on a dry matter basis has 9.5% protein. The comparison of protein content in our baby corn samples with the protein content of the aforementioned authors shows that as plants mature and their dry matter content rises, the amount of total protein in the plant decreases. These observations are consistent with those of Mader et al. [29] and Nogoy et al. [30].

Different parameters such as blanching time, temperature, CaCl_2_ concentration as pre-treatment, salt concentration, sugar concentration, and fermentation time affect the physicochemical, microbial and sensory characteristics of pickled baby corn [31]. According to the results presented in Table 1, the control baby corn sample had a significantly higher (*p* < 0.05) content of total sugars than the remaining tested baby corn samples with salt. The total sugar content in the control baby corn sample amounted to 28.64%, which was about 4.5-, 4.2- and 5-fold higher than that found in the popping, semi-flint and sweet baby corn samples with salt, respectively. The lowest total sugar content (5.87%) was recorded in the sweet baby corn sample with salt, while the percentage contribution of total sugars of popping and semi-flint baby corn samples to that in the sweet baby corn sample was higher by about 9.9% and 17.5%, respectively. Although sweet corn endosperms are known to accumulate more sugar and a highly branched-water-soluble type of starch compared to regular corn varieties, our results could be related to the findings of Azanza et al. [32], who reported that some sweet corn varieties are characterized by a rapid loss of kernel quality after harvest due to the conversion of sugars to starch and phytoglycogen and moisture loss. In addition, our results for sugar content are in accordance with results reported by other authors [33,34]. The average value for total sugar content in baby corn samples with salt and sugar was 16.81%, which was about 61% higher from that determined in the same baby corn samples with only added salt. Given that the impact of processing and various links in the food chain are some of the elements that affect the nutritional composition of products, the obtained results were as expected. Along with acting as a sweetener, sugars are also used in food as colouring, preservatives, and acidity regulators. Additionally, no other natural or artificial sweetener can match sucrose’s crucial functional qualities in a variety of food products [35]. Sucrose was the main soluble sugar, accounting for approximately 62% of the soluble sugars in the control baby corn sample (Table 1), which is less than the 66.4% and 77.1% reported by Ledenčan et al. [36]. In contrast, sucrose comprised between 9.6% and 30% of the total soluble sugar recorded in the samples of popping, semi-flint and sweet baby corn with salt and the same samples with salt and sugar (Table 1). The obtained results indicate that differences in the genetic background of the corn hybrids affected the differences in the genes responsible for controlling sugar synthesis. In support of this, the research by Zhu et al. [37] found that sucrose content accounted for approximately 94%, 88%, and 85% of the soluble sugars in the different sweet corn genotypes. Generally, the content of each sugar, as well as the total sugar content, reach maximum levels at about 17 days after pollination and decrease during maturation [36]. The sucrose content in the control baby corn sample amounted to 17.79%, which was about 29-, 10- and 16-fold higher than that found in the popping, semi-flint and sweet baby corn samples with salt, respectively. However, the data obtained points out the smaller differences in the content of sucrose between the control baby corn sample and the tested baby corn samples with salt and sugar. The sweet baby corn sample with salt and sugar showed a high content of sucrose—5.82%. In contrast, a 43% and 40% lower concentration of sucrose was detected for the popping and semi-flint baby corn samples with salt and sugar, respectively. As expected, the tested baby corn samples with the salt and sugar addition had a 1.8- to 5.2-fold higher content of sucrose than the same baby corn samples with only salt added. However, all sugars should be taken into account when determining sweetness, because they can have varying levels of flavour intensity [38]. For that reason, the content of total reducing sugars was also determined. The content of total reducing sugars ranged from 4.81% to 5.80% and from 11.09% to 13.13%, having significant difference between baby corn samples with salt and with the salt and sugar addition (*p* > 0.05). Among the tested baby corn samples with only the salt addition, the highest total reducing sugar content was detected in popping baby corn, followed by semi-flint baby corn and sweet baby corn, having a value of 5.80%, 5.00% and 4.81%, respectively. The average value of these three baby corn samples for total reducing sugars was 5.20%, which was about 138% lower from that determined in the baby corn samples with the salt and sugar addition. Additionally, in the control baby corn sample, the content of total reducing sugars amounted to 9.91%.

The nutritional quality characteristics of the tested baby corn samples, such as the content of dietary fibres (NDF, ADF, ADL, hemicellulose, and cellulose), were also determined in this study and presented in Table 1. One of the significant characteristics of all baby corn samples enriched with bioactive compounds from BSSC is the high content of lignocellulosic fibres (Table 1), which affects the improvement of their nutritional and technological quality. The highest content of NDF was detected in the control baby corn sample (46.21%), while the popping, semi-flint and sweet baby corn samples with salt were found to have a 24.7%, 30.9% and 32.9% lower content of NDF, respectively. Among the tested baby corn samples with the salt and sugar addition, the lowest NDF content (26.46%) was found in the sweet baby corn sample, and no significant differences were found between these three baby corn samples (*p* > 0.05). The content of ADF in the control baby corn amounted to 13.05%, which was about 2-fold higher than that found in the popping, semi-flint and sweet baby corn samples with salt. Furthermore, the control baby corn sample had a 51%, 50.2% and 56.9% higher content of ADF compared to the popping, semi-flint and sweet Baby corn, respectively. Statistically significant differences in the ADF content were not observed between samples with added salt and those with added salt and sugar. However, significantly high variations in NDF (20.33%) and ADF (34.07%) as well as ADL (58.41%) contents were noticed among the tested baby corn samples (Table 1). These differences indicate the genotypic variations for lignocellulosic fibres, even within such a small set of corn hybrids. Additionally, this suggested that the fibres’ solubility and physiochemical properties changed during processing. ADL is the most complex of the three-dimensional amorphous biopolymers, encapsulating the cellulose and hemicellulose through hydrogen and covalent bonds [39,40,41]. The content of ADL in the observed baby corn samples ranged from 1.68% (flint baby corn) to 6.46% (control baby corn). The obtained range of the tested baby corn samples was higher as compared with the ADL content values of grain for four different corn genotypes [42], but it was lower than those reported for corn cobs, which were in the range of 6.73−13.9% [43]. The comparison of our results for ADL content with the results of the aforementioned authors shows that the observed differences in ADL content could be attributed primarily to the fact that the consumption of baby corn products includes both the cob and the kernel. In addition, among tested baby corn samples, the control baby corn sample contained the highest content of NDF, ADF, and ADL. The observed differences in the content of lignocellulosic fibres could be attributed to different harvest dates of the control baby corn and popping, semi-flint, and sweet ones, since with maturity and an increasing dry matter content in corn cob, the content of the ADF and NDF in other parts of the corn plant increases [44]. Cellulose and hemicellulose, which are principal non-starch polysaccharides present in corn grain [27] ranged from 3.06% (sweet baby corn sample with salt and sugar) to 6.56% (control baby corn sample) and from 20.20% (semi flint with salt and sugar) to 33.16% (control baby corn sample), respectively. The control baby corn sample had the highest content of cellulose, while the popping, semi-flint and sweet baby corn samples were found to have on average a 35.5%, 28.7% and 44.6% lower content of cellulose, respectively. Nevertheless, no significant differences were found between the different baby corn samples with salt and the baby corn samples with salt and sugar additions. It is worth underlining that the observed statistically significant differences in the content of cellulose and hemicellulose between the control baby corn sample and the other tested baby corn samples can be attributed to their biodegradability and the influence of the pH values of the solution on their dissolution. Our results are well in accordance with data reported by Yang et al. [45], who showed that the dissolution of hemicellulose at different pH values was in the following descending order: pH4 > pH3 > pH5. The precise assessment of the chemical composition of different baby corn samples is very important for the evaluation of their suitability for the preparation of functional products with health-promoting potentials and high nutritive value.

### 2.2. Phenolic Compound Content, the Antioxidant Capacity and the Colour of Baby Corn Samples

Studies have shown that corn represents a good dietary source of polyphenols, strong antioxidants and regulators of human immune system associated with lowering the risk of various degenerative diseases, prevention of cardiovascular diseases, cancer and age-related diseases [46]. The content of phenolic compounds, phenolic acids, flavonoids, anthocyanins and antioxidant capacity of the baby corn samples are given in Table 2. In order to evaluate the functional advantages of anthocyanin-enriched pickled baby corn samples, all the results were compared with those of commercial baby corn product as a control. The content of total phenolic compounds in the tested baby corn samples ranged from 3540.54 to 6534.21 mg GAE/kg. Zhang et al. [47] tested eight different sweet corn genotypes and reported that the total phenolic contents ranged from 3800 to 4776 mg GAE/kg, which is higher than that in our control sample (3540.54 mg GAE/kg). However, in a study by Song et al. [48], the free phenolic content in sweet corn was as low as 2670 mg GAE/kg. On average, the enriched popping baby corn samples had about a two-fold higher content of total phenolic compounds compared to that in the control baby corn sample. This was expected, since the addition of BSSCs in the baby corn samples had a positive impact on phenolic compound content (Table 2). Among the tested baby corn samples, the highest total phenolic content was detected in popping baby corn with a salt addition, followed by the semi-flint and sweet baby corn samples, which had a 10.8% and 16.3% lower content than in the popping one, respectively. No significant differences were found between the baby corn samples with salt and the salt and sugar addition (*p* > 0.05) among the same corn genotype.

Phenolic acids were also characterized in the baby corn samples, and the contents are given in Table 2. The predominant phenolic acid present in the popping, semi-flint and control baby corn samples was chlorogenic acid, which ranged from 542.91 to 1558.57 µg/g d.m. On the contrary, the sweet baby corn sample had the lowest content of chlorogenic acid content, with a value of 15.50 µg/g d.m. These differences suggest that even among a small group of corn hybrids, there are genotypic variations for phenolic acids. Chlorogenic acid, 3,4-dihydroxybenzoic acid and ferulic acid were identified as the main phenolic acids in the tested baby corn samples. Our results are well in accordance with data reported for sweet corn genotypes [49] as well as with literature data reporting that *p*-coumaric acid and ferulic acid were commonly identified in cereals [50,51,52]. Additionally, it had been reported that quercetin, kaempferol, vanillic, caffeic, and gallic acid were present [53,54]. The popping baby corn samples with a salt addition had a 33.3% and a 55.6% higher content of 3,4-dihydroxybenzoic acid compared to the semi-flint and sweet baby corn samples, respectively. In addition, the semi-flint and sweet baby corn samples with added salt and sugar also had a lower content of 3,4-dihydroxybenzoic acid (103.21 and 77.39 µg/g d.m., respectively) than that of the popping one (137.23 µg/g d.m.). Statistically significant differences were observed in 3,4-dihydroxybenzoic acid content between samples with added salt and those with added salt and sugar. Furthermore, the ferulic acid content of the tested baby corn samples ranged from 34.90 g/g d.m. up to 232.53 μg/g d.m. (Table 3). However, our results for free ferulic acid are much lower than those reported by De la Parra et al. [55] for corn kernels (495–970 μg/g), but similar when compared to the free ferulic acid content values of QPM, baby, popping and sweet corn genotypes grown in India [3]. Findings reported by Saulnier and Thibault [56] showed that ester bonds hold the majority of the total ferulic acid to the components of the cell wall, and they also showed that approximately 76−91% of the total ferulic acid in corn kernels are in the bound form [52]. The highest content of ferulic acid (232.53 μg/g d.m.) was recorded in the control baby corn sample, while average percentage contribution of ferulic acid of the popping, semi-flint and sweet baby corn samples compared to that in the control was lower by about 49%, 83% and 78%, respectively. Additionally, in the control baby corn sample, two individual phenolic acids, namely vanillic (66.94 μg/g d.m.) and syringic (7.79 μg/g d.m.) acid, were detected.

The total flavonoid content varied greatly among the tested baby corn samples, and it ranged from 2.74 to 4.06 mg/kg d.m. The control baby corn sample had the lowest total flavonoid content (2.74 mg/kg d.m.) compared to those of the popping and semi-flint genotypes (3.99 and 3.55 mg/kg d.m., respectively) as well as the sweet baby corn sample (2.95 mg/kg d.m.) (Table 2). The content of rutin was highest in the popping baby corn sample with the salt addition, and it amounted to 180.21 μg/g d.m., which was 1.45-fold and 1.43-fold higher than that found in the semi-flint and sweet baby corn samples, respectively. Rutin, as the most efficient copigment [57], has not been detected in control baby corn samples. It is noteworthy that rutin has biological effects, which include a decrease in post-thrombotic syndrome, endothelial dysfunction, or venous insufficiency [58,59], which suggests great functionality and added value for the tested baby corn samples compared to the control baby corn. Epicatechin originating from BSSC was also quantified (Table 2). The contents of epicatechin in the tested baby corn samples ranged from 148.23 to 167.39 μg/g d.m.

As we expected, the addition of BSSC had a positive impact on anthocyanin content in the baby corn samples, as well as on its colour (Table 2, Figure 1). In complete contrast to the control baby corn sample, the popping, semi-flint and sweet baby corn samples were rich in anthocyanins. Furthermore, the data obtained points out the significant difference in the content of anthocyanins between popping and sweet baby corn samples with the salt addition and the salt and sugar addition (Table 3). In this study, we observed three anthocyanins in the baby corn samples. The most prevalent anthocyanin in the popping, semi-flint and sweet baby corn samples was cyanidin 3-glucoside (Cy-3-Glu). The content of Cy-3-Glu ranged from 184.62 to 247.47 μg/g d.m., while the antioxidant activity of these baby corn samples amounted to about 48.79 to 54.54 mmol Trolox Eq/kg d.m. Kähkönen and Heinonen [60] found that the health and therapeutic effects of anthocyanins are related to their chemical and biochemical reactivity, which are partially explained by their antioxidant activities. The in vitro antioxidant activity of anthocyanidins in preventing the oxidation of human LDL decreases in the following order: delphinidin = cyanidin > malvidin > peonidin > pelargonidin > petunidin, while glycosylation alters the activity order. The content of Cy-3-Glu was followed by De-3-Glu, which ranged from 12.24 to 34.93 μg/g d.m., and Pg-3-Glu, which ranged from 12.50 to 17.61 μg/g d.m. (Table 3). According to the results of Kim et al. [61], Cy-3-Glu was the most common anthocyanin in the whole-grain flour of purple barley (214.8 µg/g), while De-3-Glu was the most abundant anthocyanin in the whole-grain flour of blue (167.6 µg/g) and black (36.0 µg/g) barley. In our study, the content of the second major defined anthocyanin De-3-Glu was 12-fold and 15-fold lower than that of Cy-3-Glu in the salt and salt–sugar popping baby corn samples, respectively. The same trend was observed in the sweet baby corn samples, with a 7-fold and 10-fold higher content of cyanidin 3-glucoside. In the control sample of baby corn, anthocyanins were not detected, which additionally indicates the functionality and added value of the tested baby corn samples. Overall, in our study, the control baby corn sample had the lowest content of total phenolics, flavonoids, anthocyanins, and ABTS radical scavenging activity (40.48 mmol Trolox Eq/kg d.m.).

The addition of anthocyanin-rich BSSCs changed the baby corn cobs’ colour to a red range. The baby corn samples had CIE a* values that were more than 16-fold higher than that of the control baby corn sample (Table 3, Figure 1). According to previous reports, pelargonidin 3-glucoside has a lesser red-range colouring capability than cyanidin 3-glucoside [62]. In addition, the control baby corn sample had yellow colour values (CIE b*) higher by 86.3%, 79.8% and 85.0% than the popping, semi-flint and sweet baby corn samples with salt and sugar, respectively. Typically, the colours of anthocyanin vary from red at low pH to blue and green at high pH [63]. Considering the investigation of Heredia et al. [64], the red flavylium cation should be dominant in our samples since baby corn samples have a very acidic pH (pH 1–3). The basic difference in the cob colour of the three baby corn samples enriched with black soybean coat was in the CIE b* values. Differences in colour between individual samples occur as a consequence of the difference in the composition of anthocyanins, which, depending on the aglycone and sugar components, exhibit different bioactivity and colour. The predominant content of cyanidin-based glucosides undoubtably contributed to the more reddish-purple colour of the investigated baby corn samples. As cyanidin reflects in the red wavelengths of the spectrum (512 nm) [65], its predominance would lead to a more reddish hue. However, the diacylated cyanidin derivative even displayed a blue appearance [66]. As contributors to the colour of cereals, pelargonidin and delphinidin appear as an orange–red and blue–purple pigment, respectively [67].

### 2.3. Sensory Properties of Baby Corn Samples

Sensory analysis is a valuable tool in new food product development and market placement and is associated with food acceptability. It is a useful tool in product development, product improvement and quality maintenance [68]. Simić et al. [69] have reported that whole-grain coloured corn flour affected bread flavour, providing a pleasant odour and taste and a very intensive flavour, although numerous phenolic compounds with important health benefits are characterized by bitterness and astringency [70]. A salty brine is generally used because salt enhances the consistency and flavour of vegetables [71,72] while maintaining a crisp texture [73]. Salt inhibits the development of microorganisms that produce the pectin-digesting enzymes that soften pickled vegetables. Controlling the salt concentration in the early stages of fermentation will ensure the desired quantity of acid generation.

Descriptive sensory analysis by the Flash Profile method was applied to defined and compare the sensory profiles of the produced samples of pasteurized baby corn cobs enriched with anthocyanins and the commercially available baby corn cob products in salted brine (Figure 2). The obtained results were subjected to the Generalized Procrustes Analysis and are shown on two bi-plot diagrams (Figure 2a,b), with the display of the first three main components for a better understanding of the results (˃83% of the variability of the results). The diagrams clearly show the differences between the samples that arose as a result of the varietal specificity of the samples. The commercially available product (sample G) differs from the other samples, having the more pronounced and noticeable odour and flavour of cooked corn, as well as a more uniform colour. In contrast to this sample, samples A and B had a clearly expressed fermented flavour; moreover, sample A as well as samples C and D possess flavour associated with plant mold. Chewiness is the number of chews required to disintegrate the sample and bring it to a state suitable for swallowing. Samples B, E and F expressed higher chewiness compared to the other analysed samples. The addition of sugar to the marinating brine, in addition to the salt, contributed to sample B being perceived as noticeably less sour compared to sample A and with a more pleasant taste The acidity of this product was marked as the most acceptable (least acidic), which was previously confirmed by the chemical analysis of the pH value (Table 1).

## 3. Materials and Methods

### 3.1. Plant Material

The plant material encompassed three different corn hybrids recently developed at the Maize Research Institute, Zemun Polje, Serbia (MRIZP), namely a sweet corn ZP 553*su* (normal sweet corn of *su* genotype, female parent obtained from the population with enhanced sugar content, and male parent from the F2 population), a popping corn ZP 6119k (FAO 600, Female parent from the F2 population of the Supergold origin, male parent from the South American popcorn germplasm), and a semi-flint genotype ZP 161 (FAO 200, parents from local collection MRIZP), as well as the soybean variety Black Tokyo (plant introduction maintained in MRIZP soybean collection, originated from Asia). All genotypes were grown in the experimental field in the vicinity of Belgrade (44°52′ N, 20°19′ E, 82 m a.s.l.), Serbia, in the 2021 growing season. Standard cropping practices were applied to provide adequate nutrition and to keep the plots disease- and weed-free. Baby corn ears were hand harvested one day after silk emergence (before pollination) successively for a period of seven days and transferred to the laboratory. The black soybean was harvested at a stage of full maturity. Soybean kernels were manually dehulled, and seed coat was ground in a laboratory mill (Perten 120 lab mill, Perten, Hägersten, Sweden) to fine powder (particle size < 500 μm) and stored at −70 °C prior to the extraction of anthocyanins. Based on our previous study [17], black soybean seed coat contained 40,762.2 mg GAE/kg of total phenolic compounds and 11,883.90 mg GGE/kg of total anthocyanins, with cyanidin 3-glucoside as the dominant (3387.10 µg/g). The antioxidant capacity of black soybean seed coat determined by QUENCHER method was 399.5 mmol Trolox Eq/kg.

### 3.2. Technological Steps for Baby Corn Preparation

#### 3.2.1. Anthocyanins Extraction from BSSC

Extraction of anthocyanins from the ground BSSC was performed (in a ratio 1:40 *w*/*v*) with acetic acid (0.8%) and the addition of lactic acid (1%) as an anthocyanin stabilizer. Extraction while heating at 50 °C was conducted on a magnetic stirrer (IKA^®^ C-MAG HS 7 hotplate stirrer, IKA^®^-Werke GmbH & Co. KG, Staufen im Breisgau, Germany) for 60 min, as described previously by Nikolić et al. [11]. After centrifuging (Dynamica Velocity 18R Refrigerated Benchtop Centrifuge, DKSH New Zealand Limited, Palmerston North, New Zealand) for five minutes at 11,000 rpm (12,175× *g*), the clear supernatant was used to marinate the brine preparation. Using a spectrophotometer (Agilent 8453 UV-visible Spectroscopy System, Agilent, Santa Clara, CA, USA), the supernatant’s absorbance was measured at 535 and 700 nm in order to calculate the content of anthocyanins. These anthocyanins were then quantified and expressed as mg of cyanidin 3-glucoside equivalent (CGE) per kg of dry matter (d.m.) [74]. The total content of anthocyanins quantified in extract amounted to 4000 mg CGE/kg d.m., and it was used as basic marinating brine.

#### 3.2.2. Marinating Brine Preparation

The total anthocyanin content in the obtained extract that was used as basic marinating brine amounted to 4000 mg CGE/kg. Two marinating brine recipes were used in the experiments: 1. Basic marinating brine with the addition of 3% of table salt; 2. Basic marinating brine with the addition of 3% of table salt and 2% of caster sugar.

#### 3.2.3. Preparation of Baby Corn Products

After dehusking and removing the silk from the corn ears of three different ZP corn hybrids, the baby corn cobs were measured, sorted and graded according to length and diameter. Cleaned corn cobs were blanched in hot water at 95 °C for 120 s (shaking water bath GFL 1083, GFL mbH, Berlin, Germany), cooled and submerged in a brine solution. Sterile glass jars (2500 mL) were filled with baby corn cobs. In order to enrich baby corn with anthocyanins, the cobs were first immersed in the prepared extract of BSSCs and kept in a dark place at room temperature for 7 days. After that, the baby corn cobs were transferred into smaller, previously sterilized jars (350 mL), filled with brine 1 or brine 2 liquid, closed and pasteurized. Three jars of each baby corn hybrid were prepared. All baby corn products were kept in a dark place at room temperature, and analyses of chemical composition and sensory characteristics were performed after three months to determine product quality. The sensory qualities of the pickled baby corn samples were compared to a commercial baby corn product bought at a local supermarket as a control. According to the declaration, the only ingredients in the solution used to produce commercial baby corn were citric acid and salt. For chemical composition evaluation, the pickled baby corn samples were dried in a ventilation oven (Memmert UF 55, Memmert GmbH + Co. KG, Schwabach, Germany) for 48 h at 60 °C and ground on a laboratory mill (Perten Instruments, Hägersten, Sweden) (mesh 0.5 mm).

### 3.3. Chemical Procedure

#### 3.3.1. Analysis of Total Protein Content in Baby Corn

The protein content was determined by the standard micro-Kjeldahl method as the total N multiplied by 6.25 [75] on BÜCHI Kjeldahl System (Auto Kjeldahl Distilation Unit K-350 and Speed Digester K-439, BÜCHI Labortechnik, Flawil, Switzerland). Results were expressed in percentages per dry matter (d.m.).

#### 3.3.2. Analysis of Sugars Content in Baby Corn

The content of total sugars, reducing sugars and sucrose was determined by the Luff-Schoorl method [76]. The Luff-Schoorl method is based on the reaction between reducing sugars and alkaline solution of copper sulphate, with a subsequent reduction of cupric copper to cuprous oxide. In the method, Cu^2+^ ions that had not been reduced are determined iodometrically. Furthermore, total sugars were then determined by converting nonreducing sugars into reducing sugars through acid hydrolysis. The percentage of sucrose was calculated as the difference between total and reducing sugars, i.e., as the difference between total inverter and natural inverter. All the results are given as the percent per d.m.

#### 3.3.3. Analysis of Dietary Fibres Content in Baby Corn

The content of hemicellulose, cellulose, neutral detergent fibres (NDF), acid detergent fibres (ADF), and acid detergent lignin (ADL) were determined by the Van Soest detergent method modified by Mertens [77] using the Fibertec system FOSS 2010 Hot Extractor (FOSS Tecator, Hoeganaes, Sweden). The method is based on the fibres’ solubility in neutral, acid, and alkali reagents. NDF practically represents total insoluble fibres (not soluble in water), and ADF mainly consists of cellulose and lignin, while ADL is pure lignin. NDF was measured after boiling sample (1 g) in 100 mL of a special detergent under a neutral (pH 7) condition over 60 min and filtering. The liquid that passed through the sintered disc filter contained starch, sugar, protein and other compounds that were dissolved. ADF was determined in much the same way, except that a different detergent was used under acid (pH 2) conditions. Because of the different detergent and acid conditions, hemicellulose and cell solubles were dissolved and filtered away. ADL was measured by further treating ADF with strong acid (72% H_2_SO_4_), which dissolved cellulose. After filtering and drying, the NDF, ADF and ADL were calculated as a percentage of the original sample. The content of hemicellulose was obtained as the difference between NDF and ADF contents, while the cellulose content was calculated as the difference between ADF and lignin contents. All the results are given as the percent per d.m.

#### 3.3.4. Extraction of Free Soluble Phenolic Compounds from Baby Corn

Extracts were prepared by continuous shaking of 0.5 g of sample in 10 mL of 70% (*v*/*v*) acetone for 30 min at room temperature. After centrifugation at 10,000 rpm (10,062× *g*) for 5 min, a supernatant was used for the detection of total phenolics and total flavonoids. For the detection of free soluble phenolic acids and flavonoids by HPLC, 5 mL of extracts were evaporated under N_2_ stream at 30 °C to dryness (Reacti-Therm nitrogen evaporator system 18821, Thermo Fisher Scientific Inc., Waltham, MA, USA). Final residues were redissolved in 1.2 mL of methanol. The extracts were kept at −70 °C prior to analysis.

#### 3.3.5. Analysis of Total Free Soluble Phenolic Content in Baby Corn (TPC)

The total phenolic content was determined according to the Folin–Ciocalteu procedure [78]. The extract (50 µL) was transferred into a test tube and the volume was adjusted to 500 µL with distilled water and oxidised with the addition of 250 µL Folin–Ciocalteu reagent. After 5 min, the mixture was neutralised with 1.25 mL of 20% aqueous Na_2_CO_3_ solution. After 40 min, the absorbance was measured at 725 nm. The total phenolic content was expressed as mg of catechin equivalent (CE) per kg of d.m.

#### 3.3.6. Analysis of Total Flavonoid Content in Baby Corn (TFC)

Extracts were prepared by continuously shaking 0.5 g of baby corn samples in 10 mL of 70% (*v*/*v*) acetone for 30 min at room temperature. After centrifugation at 5000 rpm (2516× *g*) for 5 min, spectrophotometric measurements at 360 nm (ε = 13.6 mM^−1^ cm^−1^) were used to determine total flavonoids in the extract [79]. The content is expressed as mg per kg of d.m.

#### 3.3.7. Analysis of Total Anthocyanin Content in Baby Corn (TAC)

Anthocyanins were extracted from 50 mg of baby corn samples by mixing with 5 mL of methanol acidified with 1 M HCl (85:15, *v*/*v*). After shaking on a horizontal shaker (MLW Thys 2, VEB MLW Labortechnik, Ilmenau, Germany), the absorbance was measured on a spectrophotometer at 535 and 700 nm [74]. The content was calculated using the molar extinction coefficient of 25,965 Abs/M·cm and a molecular weight of 449.2 g/mol and expressed as mg of cyanidin 3-glucoside equivalent (CGE) per kg of d.m.

#### 3.3.8. Analysis of Individual Anthocyanins in Baby Corn by HPLC

Individual anthocyanins were determined from the prepared extracts after their filtration through the nylon syringe filter of 0.45 μm. The HPLC analysis was carried out with the HPLC-DAD system (Thermo Scientific Ultimate 3000) using Thermo Scientific Hypersil GOLD aQ C18 column (150 mm × 4.6 mm, i.d., 3 µm) for separation according to the method described by Žilić et al. [16]. Chromatograms were obtained at 530 nm after injection of 10 μL of sample. A linear gradient elution program, as described by Žilić et al. [16], was used with a mobile phase containing solvent A (formic acid/H_2_O, 1:99, *v*/*v*), and solvent B (formic acid/acetonitrile, 1:99, *v*/*v*) at a flow rate of 0.7 mL/min and a column oven temperature of 30 °C. Pure anthocyanin compounds such as delfinidin-3-glucoside (De-3-Glu), cyanidin-3,5-diglucoside (Cy-3,5-diGlu), cyanidin-3-sophoroside (Cy-3-Sop), cyanidin-3-glucoside (Cy-3-Glu), cyanidin-3-rutinoside (Cy-3-Rut), petunidin-3-glucoside (Pt-3-Glu), pelargonidin-3-glucoside (Pg-3-Glu) and malvidin-3-glucoside (Mv-3-Glu) were used as standards. The identified anthocyanin peaks were confirmed and quantified using the Thermo Scientific Dionex Chromeleon 7.2. chromatographic software, and the results are expressed as µg per g of d.m.

#### 3.3.9. Analysis of Individual Phenolic Acids and Flavonoids in Baby Corn by HPLC

Chromatographic analyses were performed on the Thermo Scientific Ultimate 3000 HPLC with a photodiode array detector (Thermo Fisher Scientific Inc., Waltham, MA, USA). Phenolic acids were separated on the Thermo Scientific Hypersil GOLD aQ C18 column (150 mm × 4.6 mm, i.d., 3 µm) using a linear gradient elution program with a mobile phase containing solvent A (formic acid/HO, 1:99, *v*/*v*) and solvent B (methanol) at a flow rate of 0.8 mL/min. The solvent gradient was programmed as described by Žilić et al. [52]. The chromatograms were recorded at 280 nm by monitoring spectra within the wavelength range of 190–400 nm. Identified phenolic acid peaks were confirmed and quantified using the Thermo Scientific Dionex Chromeleon 7.2. chromatographic software. Standards of galic acid, 3,4-dihydroxybenzoic acid, chlorogenic acid, caffeic acid, *p*-coumaric acid, sinapic acid, syringic acid, ferulic acid and isoferulic acid, as well as catechin, rutin and quercetin were used.

#### 3.3.10. Analysis of the Total Antioxidant Capacity of Baby Corn (AC)

The total antioxidant capacity was measured according to the direct or QUENCHER method described by Serpen et al. [80] using the ABTS (2,2′-azino-bis(3-ethilbenzothiazoline-6-sulfonic acid)) reagent. The total antioxidant capacity was expressed as mmol Trolox equivalents (Eq) per kg of d.m.

### 3.4. Measurement of Colour of Baby Corn

Colour properties of Baby corn samples were measured instrumentally by using Konica Minolta colorimeter (CR-400/410, Konica, Minolta, Tokyo, Japan), which was calibrated against a white calibration standard (CM-A70). The surface colours of samples were given as average L*—brightness (from 0 (black) to 100 (white)), a*—greenness/redness (from a* < 0 (green) to a* > 0 (red)), b*—blueness/yellowness (from b* < 0 (blue) to b* > (yellow)) values of four measurements per samples.

### 3.5. Baby Corn Product Sensory Evaluation

A panel of 10 trained sensory panelists (6 women and 4 men, aged from 37 to 51 years) was recruited to perform Flash Profile analysis. All panelists were recruited from the Maize Research Institute Zemun Polje, Belgrade-Zemun, Serbia. The procedure of Flash Profile consisted of two sessions with an intersession between them. In both sessions, samples were presented to panelists simultaneously. First session was devoted to the creation of list of attributes, arranged according to sense modality (appearance, odour, taste, flavour, texture, mouthfeel, and sound), with the idea that chosen attributes contribute highly to differentiation of samples. The panelists had to put on the list as many descriptive attributes as they wanted, but they had to avoid hedonic terms. The process of attribute generation lasted around 40–60 min. Intersession is designed in such a way that panelists exchange their lists of perceived attributes through discussion, and it enables them to update their own list if they want. The panellists were allowed to add new terms or to replace their own terms with those more appropriate. In the second session, the assessors had to rank samples in order of perceived level of intensity, from “low” to “high”, on each of the attributes from their list. For the samples with the same perceived intensity, they were allowed to share the same rank number.

### 3.6. Statistical Analysis

The data were reported as mean ± standard deviation of at least three independent repetitions. The results were statistically analysed using the Statistica software version 5.0 (StatSoft Co., Tulsa, OK, USA). Significance of differences between samples was analysed by the Tukey’s test. Differences between the means with probability *p* < 0.05 were considered as statistically significant. The results obtained within Flash Profile analysis were analysed by using multivariate statistical technique General Procrustes Analysis (GPA), and statistical analysis was performed by using XLSTAT software (version 2019.3.2, Addinsoft).

## 4. Conclusions

The enrichment of the marinating brine with anthocyanins extracted from the BSSC improved the functional profiles of the pickled baby corn products, increasing the content of total phenolic compounds, anthocyanins, and phenolic acids. The content of total anthocyanins in the tested baby corn samples ranged from 748.55 to 881.23 mg CGE/kg d.m., while in the control baby corn sample, anthocyanins were not detected. The popping baby corn sample with the highest total phenolic, flavonoid and anthocyanin content had the highest antioxidant capacity (54.54 mmol Trolox Eq/kg d.m.), which made it the most suitable for this type of product. One of the significant characteristics of all baby corn samples enriched with bioactive compounds from BSSCs was the high content of lignocellulosic fibres, which affects the improvement of their nutritional and technological quality. Results of the sensory evaluation revealed clear differences between the samples that arose as a result of the varietal specificity of samples.

## Figures and Tables

**Figure 1 plants-12-01812-f001:**
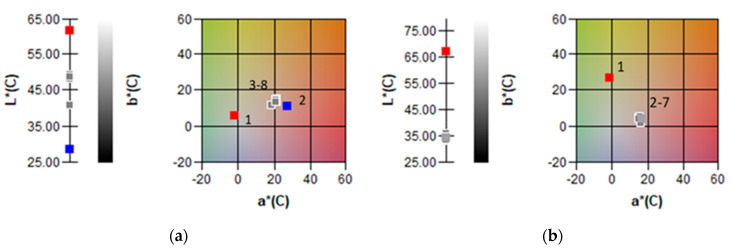
CIE a*, b* and L* chromaticity diagram for: (**a**) 1—control—brine of control sample, 2—marinating brine; 3–8—brine of samples after 3 months; (**b**) 1—control—control cob; 2–7—Baby corn cobs of all samples.

**Figure 2 plants-12-01812-f002:**
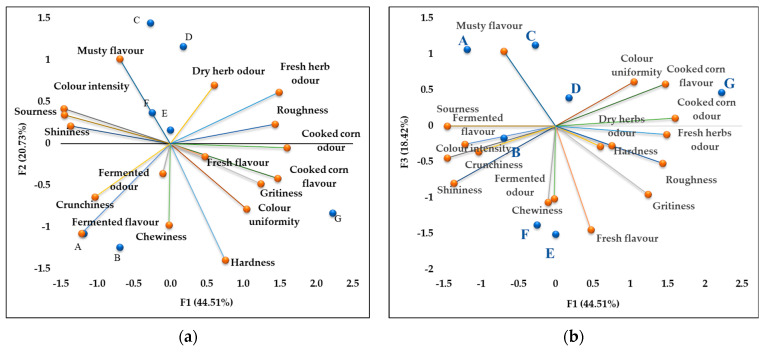
Descriptive sensory analysis: bi-plot diagrams obtained by the Flash Profile method. (**a**) Main components F1 and F2; (**b**) main components F1 and F3. A—Popping corn salt, B—Popping corn salt + sugar, C—Semi flint salt, D—Semi flint salt + sugar, E—Sweet corn salt + sugar, F—Sweet corn salt, G—Control corn.

**Table 1 plants-12-01812-t001:** Content of protein, sugars, dietary fibres and pH values in baby corn samples (% d.m.).

Compounds	Control	Popping Corn		Semi Flint Corn		Sweet Corn	
		Salt	Salt + Sugar	Salt	Salt + Sugar	Salt	Salt + Sugar
Protein (%)	17.86 ± 0.07 ^a^	17.21 ± 0.07 ^b^	16.61 ± 0.01 ^c^	14.61 ± 0.13 ^e^	12.89 ± 0.01 ^f^	17.08 ± 0.12 ^bc^	15.29 ± 0.29 ^d^
Sugars							
Total sugars (%)	28.64 ± 0.72 ^a^	6.45 ± 0.18 ^e^	14.73 ± 0.06 ^b^	6.90 ± 0.76 ^de^	16.47 ± 0.11 ^a^	5.87 ± 0.22 ^e^	19.24 ± 0.37 ^a^
Sucrose (%)	17.79 ± 0.05 ^a^	0.62 ± 0.02 ^d^	3.45 ± 0.10 ^c^	1.80 ± 0.07 ^cd^	3.31 ± 0.13 ^c^	1.11 ± 0.08 ^d^	5.82 ± 0.17 ^b^
Total reducing sugars (%)	9.91 ± 0.07 ^c^	5.80 ± 0.25 ^d^	11.09 ± 0.11 ^b^	5.00 ± 0.15 ^e^	12.99 ± 0.42 ^a^	4.81 ± 0.16 ^e^	13.13 ± 0.16 ^a^
Dietary fibres							
NDF (%)	46.21 ± 0.08 ^a^	34.80 ± 0.57 ^b^	32.09 ± 0.81 ^c^	31.95 ± 0.73 ^c^	26.69 ± 0.83 ^d^	31.00 ± 0.33 ^c^	26.46 ± 0.59 ^d^
ADF (%)	13.05 ± 0.21 ^a^	6.44 ± 0.26 ^b^	6.39 ± 0.28 ^b^	6.66 ± 0.05 ^b^	6.50 ± 0.04 ^b^	6.41 ± 0.03 ^b^	5.63 ± 0.33 ^b^
ADL (%)	6.46 ± 0.44 ^a^	2.46 ± 0.12 ^bc^	1.89 ± 0.06 ^c^	1.76 ± 0.08 ^c^	1.68 ± 0.47 ^c^	2.23 ± 0.05 ^bc^	3.07 ± 0.21 ^b^
Hemicellulose (%)	33.16 ± 0.28 ^a^	28.36 ± 0.31 ^b^	25.70 ± 0.53 ^c^	25.29 ± 0.68 ^c^	20.20 ± 0.80 ^c^	24.59 ± 0.30 ^bc^	20.33 ± 0.92 ^b^
Cellulose (%)	6.59 ± 0.64 ^a^	3.98 ± 0.38 ^bc^	4.50 ± 0.23 ^b^	4.90 ± 0.13 ^b^	4.50 ± 0.50 ^b^	4.19 ± 0.08 ^bc^	3.06 ± 0.11 ^c^
pH	5.09 ^a^	3.07 ^b^	3.10 ^b^	3.03 ^c^	2.98 ^d^	3.00 ^cd^	3.01 ^cd^

NDF—neutral detergent fibres, ADF—acid detergent fibres, ADL—acid detergent lignin. Values are means of three determinations ± standard deviation. Means followed by the same letter within the same row are not significantly different according to Tukey’s test (*p* > 0.05).

**Table 2 plants-12-01812-t002:** The content of phenolic compounds, phenolic acids, flavonoids, anthocyanins and antioxidant capacity of baby corn samples (d.m. basis).

	Control	Popping Corn		Semi Flint Corn		Sweet Corn	
		Salt	Salt + Sugar	Salt	Salt + Sugar	Salt	Salt + Sugar
Total-free soluble phenolic compounds							
Total phenolics (mg CE/kg)	3540.54 ± 34 ^e^	6534.21 ± 98 ^a^	6156.57 ± 100 ^ab^	5828.93 ± 24 ^bc^	5631.75 ± 42 ^c^	5466.68 ± 40 ^cd^	4813.93 ± 202 ^d^
Total flavonoids (mg/kg)	2.74 ± 0.01 ^f^	4.06 ± 0.04 ^a^	3.91 ± 0.03 ^b^	3.73 ± 0.01 ^c^	3.36 ± 0.01 ^d^	2.98 ± 0.04 ^e^	2.92 ± 0.03 ^e^
Total anthocyanins (mg CGE/kg)	n.d.	881.23 ± 8.4 ^a^	748.55 ± 8.8 ^b^	765.24 ± 7.7 ^b^	757.76 ± 12.9 ^b^	877.63 ± 36 ^a^	752.18 ± 15 ^b^
Free soluble phenolic acids (μg/g)							
3,4-Dihydroxybenzoic acid	n.d.	177.39 ± 2.64 ^a^	137.23 ± 6.5 ^b^	118.33 ± 1.09 ^c^	103.21 ± 0.83 ^d^	78.68 ± 1.91 ^e^	77.39 ± 4.4 ^e^
Chlorogenic acid	1195.75 ± 13.4 ^c^	1558.57 ± 50 ^a^	1478.46 ± 33 ^b^	675.41 ± 5.47 ^d^	542.91 ± 17.5 ^e^	15.20 ± 1.0 ^f^	16.02 ± 0.34 ^f^
Caffeic acid	n.d.	60.75 ± 2.08 ^a^	53.05 ± 0.68 ^b^	n.d.	n.d.	n.d.	n.d.
p-Coumaric acid	32.88 ± 1.65 ^a^	6.49 ± 0.27 ^b^	5.17 ± 0.29 ^bc^	5.19 ± 0.11 ^bc^	3.67 ± 0.0 ^c^	6.79 ± 0.65 ^b^	6.03 ± 0.21 ^bc^
Ferulic acid	232.53 ± 4.98 ^a^	151.97 ± 1.30 ^b^	84.09 ± 0.18 ^c^	45.03 ± 1.67 ^d^	34.90 ± 0.98 ^e^	52.84 ± 0.21 ^d^	47.55 ± 0.76 ^d^
Vanillic acid	66.94	n.d.	n.d.	n.d.	n.d.	n.d.	n.d.
Syringic acid	7.79	n.d.	n.d.	n.d.	n.d.	n.d.	n.d.
Flavonoids (μg/g)							
Epicatechin	n.d.	167.39 ± 1.31 ^a^	149.56 ± 0.45 ^b^	163.26 ± 5.95 ^a^	164.72 ± 2.11 ^a^	164.97 ± 0.98 ^a^	148.23 ± 0.14 ^b^
Rutin	n.d.	180.21 ± 2.06 ^a^	164.89 ± 3.32 ^b^	123.83 ± 1.00 ^cd^	118.15 ± 0.06 ^de^	125.93 ± 0.18 ^c^	113.62 ± 1.23 ^e^
Anthocyanins (μg/g)							
De-3-Glu	n.d.	19.60 ± 0.17 ^b^	12.24 ± 0.17 ^c^	n.d.	n.d.	34.93 ± 1.39 ^a^	18.39 ± 1.73 ^b^
Cy-3-Glu	n.d.	233.11 ± 1.53 ^b^	188.39 ± 2.22 ^cd^	184.62 ± 1.20 ^d^	193.16 ± 0.04 ^c^	247.47 ± 2.08 ^a^	186.24 ± 0.17 ^d^
Pg-3-Glu	n.d.	n.d.	n.d.	n.d.	n.d.	17.61 ± 0.08 ^a^	12.50 ± 0.54 ^b^
Antioxidant capacity (mmol Trolox Eq/kg)	40.48 ± 0.47 ^e^	54.54 ± 0.31 ^a^	53.26 ± 0.65 ^ab^	48.79 ± 0.50 ^d^	51.83 ± 0.30 ^b^	51.19 ± 0.0 ^bc^	49.38 ± 0.95 ^cd^

n.d.—not detected, De-3-Glu—delphinidin-3-glucoside, Cy-3-Glu—cyanidin-3-glucoside, Pg-3-Glu—pelargonidin-3-glucoside. Values are means of three determinations ± standard deviation. Means followed by the same letter within the same row are not significantly different according to Tukey’s test (*p* > 0.05).

**Table 3 plants-12-01812-t003:** The measured CIE L*a*b* colour values of the investigated samples.

	Control	Popping Corn		Semi Flint Corn		Sweet Corn	
		Salt	Salt + Sugar	Salt	Salt + Sugar	Salt	Salt + Sugar
L*	67.13 ^a^	33.55 ^de^	35.69 ^b^	33.77 ^d^	33.22 ^e^	34.81 ^c^	33.95 ^d^
a*	−1.31 ^f^	16.26 ^b^	16 ^c^	15.20 ^e^	15.54 ^d^	15.17 ^e^	16.93 ^a^
b*	26.68 ^a^	1.50 ^e^	3.66 ^d^	4.63 ^c^	5.40 ^b^	3.68 ^d^	4.00 ^d^

Means followed by the same letter within the same row are not significantly different according to Tukey’s test (*p* > 0.05).

## Data Availability

All data generated or analysed during this study are included in this published article.

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
