# Peer review of "Effect of Anthocyanin-Enriched Brine on Nutritional, Functional and Sensory Properties of Pickled Baby Corn"

_plants, 2023, doi:10.3390/plants12091812_

Round 1

Reviewer 1 Report

Dear authors,

My comments are below but also included in the manuscript.

My proposal is that the authors made a graphical abstract/design of experiment to better highlight the concept of the study.

The tables must be improved by explaining the statistical significance and what represent the values (mean ± SD, for example). Also, in Table 1 please explain the abbreviation of dietary fibres.

All values that are given in the manuscript should be specified if they refer to dry or fresh weight/matter. Differences can also occur for this reason.

Line 12, 134, 329 - please re-write. Change "having in mind" with other . For example, Considering the great potential of.../Given the great....

The conclusions are too general and could be improved by highlighting the concrete results obtained.

Reviewer 2 Report

Dear authors,

The topic of the paper is very interesting, because it concern the enrichment of the nutraceutical potential of a widely consumed cereal using a natural source of anthocyanins. In some section, the paper should be modified, but in general it should be accepted for publication after minor revision.

The material and methods paragraph should be implemented, some details concerning the experimental design and chemical analyses should be added. In the following section the suggested adding will be listed:

-          3.2.1. Anthocyanins extraction from BSSC: in line 418-420 you reported: “These anthocyanins were then quantified and expressed as mg of cyanidin 3-glucoside equivalent (CGE) per kg of dry matter (d.m.)”, but above (line 404-406) a previous work, where the total phenolic compounds and total anthocyanins were quantified, was mentioned. Which values were used to calculate the quantity of BSSC extract to add to brine in order to have the concentration of 4000 μg CGE/g of anthocyanins? The new ones or the previous work ones? Or the new ones confirmed the previous values? It should be better specified. I suggest to add a paragraph in results and discussion where the results of chemical characterization of BSSC could be reported.

-          3.2.3. Preparation of Baby corn products: how many jars/thesis were prepared?

-          Chemical procedure: how many repetition/thesis were carried out? Moreover, what about the sampling? Only a jar/thesis was used for sampling or more? It should be better specified.

Moreover:

line 415 and line 480 - “11000 rpm” and “10000 rpm “: the author should indicate the speed also as Relative Centrifugal Force (RCF) expressed in x g.

line 490: Na2CO3 it should be written in the right form (Na2CO3)

line 119: “presented in Table 1. the control Baby corn sample had a significantly higher”: the dot between Table 1 and the control should be deleted.

Best regards

Reviewer 3 Report

Dear authors and the editor,

This is an interesting work and some new findings had been summarized in abstracts and the conclusion. But in my opinion, there are some problems to be answered and improved.

Q1: There is lack of some genetic background of three different corn hybrids with profiles of figure, origins of sources, and the other agronomy traits in this study.

Q2: Some chemical expression are wrong, such as Na2CO3 in line 490, please check in the whole text and pay attention to the subscript labels of numbers.

Q3: I don't think like sentents such as "Differences at p<0.05 were considered as significant." in line 565 is better to understand and good presentation, please rewrite it.

Q4: All the data should be analyzed statistically, especially with more than  triplicate samples, please improved the methods and the result.

Round 2

Reviewer 3 Report

Most problems had been improved and I suggest to accept this MS.